# HUMAN MOTIONFORMER: TRANSFERRING HUMAN MOTIONS WITH VISION TRANSFORMERS

**Hongyu Liu**[1*]  **Xintong Han**[2*]  **Chengbin Jin**[2]  **Lihui Qian**[2]  **Huawei Wei**[3]  **Zhe Lin**[2]
**Faqiang Wang**[2]  **Haoye Dong**[5]  **Yibing Song**[4†]  **Jia Xu**[2]  **Qifeng Chen**[1†]
[1]Hong Kong University of Science and Technology  [2]Huya Inc
[3]Tencent  [4]AI[3] Institute, Fudan University  [5] Carnegie Mellon University
`hliudq@cse.ust.hk`  `yibingsong.cv@gmail.com`

## ABSTRACT

Human motion transfer aims to transfer motions from a target dynamic person to a source static one for motion synthesis. An accurate matching between the source person and the target motion in both large and subtle motion changes is vital for improving the transferred motion quality. In this paper, we propose Human MotionFormer, a hierarchical ViT framework that leverages global and local perceptions to capture large and subtle motion matching, respectively. It consists of two ViT encoders to extract input features (i.e., a target motion image and a source human image) and a ViT decoder with several cascaded blocks for feature matching and motion transfer. In each block, we set the target motion feature as Query and the source person as Key and Value, calculating the cross-attention maps to conduct a global feature matching. Further, we introduce a convolutional layer to improve the local perception after the global cross-attention computations. This matching process is implemented in both warping and generation branches to guide the motion transfer. During training, we propose a mutual learning loss to enable the co-supervision between warping and generation branches for better motion representations. Experiments show that our Human MotionFormer sets the new state-of-the-art performance both qualitatively and quantitatively. Project page: https://github.com/KumapowerLIU/Human-MotionFormer

## 1 INTRODUCTION

Human Motion Transfer, which transfers the motion from a target person's video to a source person, has grown rapidly in recent years due to its substantial entertaining applications for novel content generation Wang et al. (2019); Chan et al. (2019). For example, a dancing target automatically animates multiple static source people for efficient short video editing. Professional actions can be transferred to celebrities to produce educational, charitable, or advertising videos for a wide range of broadcasting. Bringing static people alive suits short video creation and receives growing attention on social media platforms.

During motion transfer, we expect the source person to redo the same action as the target person. To achieve this purpose, we need to establish an accurate matching between the target pose and the source person (i.e., each body part skeleton in a pose image matches its corresponding body part in a source image), and use this matching to drive the source person with target pose (i.e., if the hand skeleton in the target pose is raised, the hand in the source person should also be raised). According to the degree of difference between the target pose and the source person pose, this matching can be divided into two types: **global** and **local**. When the degree is large, there is a large motion change between the target pose and the source person, and the target pose shall match a distant region in the source image (e.g., the arm skeleton of the target pose is distant from the source man arm region in Fig. 1(b)). When the degree is small, there are only subtle motion changes, and the target pose shall match its local region in the source image (e.g., the arm skeleton of the target pose is close to the source woman arm region in Fig. 1(b)). As the human body moves non-rigidly, large and subtle

---

*X. Han and H. Liu contribute equally. †Y. Song and Q. Chen are the corresponding authors.

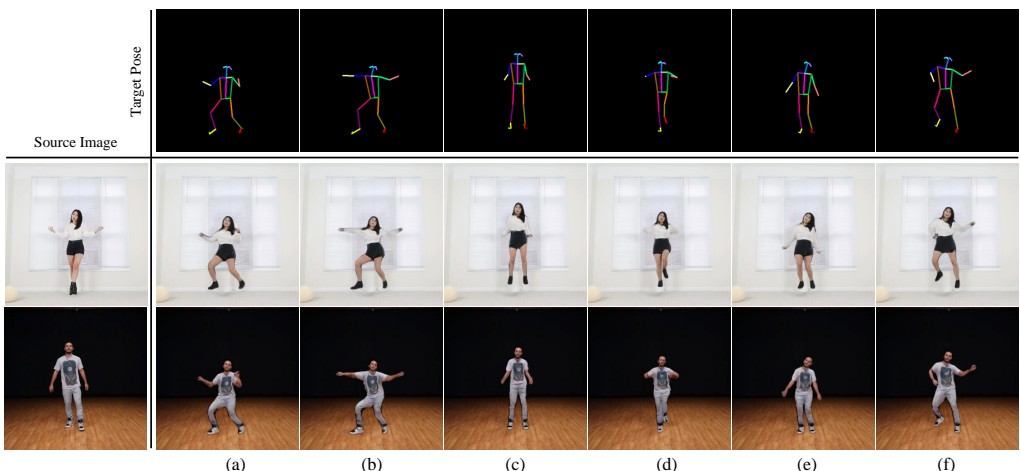

Figure 1: Human motion transfer results. Target pose images are in the first row, and two source person images are in the first column. Our MotionFormer effectively synthesizes motion transferred results whether the poses in the above two images differ significantly or not.

motion changes usually appear simultaneously. The local and global matching shall be conducted simultaneously to ensure high-quality human motion transfer.

Existing studies Ren et al. (2020; 2021); Tao et al. (2022); Zhao & Zhang (2022) leverage 2D human body keypoints Cao et al. (2017); Xiu et al. (2018) as the initial pose representations and introduce image-to-image translation Isola et al. (2017) or predictable warping fields (e.g., optical flow, affine transformations) for geometric matching. Since the 2D human body keypoints are sparse and the geometric matching is captured by CNNs, it is hard to capture global matching as the CNN receptive field is limited. As a result, artifacts occur and details are missing in the transferred results when facing large motion changes between the target pose and source person. The alternative 3D methods Neverova et al. (2018); Gafni et al. (2021); Huang et al. (2021a) introduce DensePose Alp Güler et al. (2018) or parametric body mesh Loper et al. (2015) as human representations to perform pixel-wise matching. They can globally match target pose and source images by aligning two humans into one 3D model. However, the off-the-shelf Densepose and 3D models suffer from background interference and partial occlusion, limiting the image alignment quality for dense pixel-wise matching.

***Is there a proper way to simultaneously utilize robust 2D human body keypoints and build global and local matching?*** Fortunately, recent methods Dosovitskiy et al. (2020); Liu et al. (2021c; 2022); Dong et al. (2022) demonstrate that the Vision Transformers (ViTs) can capture global dependencies in visual recognition. Inspired by this design, we combine the advantages of CNNs and Transformers and propose a Vision-Transformers-based framework named MotionFormer to model the visual correspondence between the source person image and the target pose image. As shown in Figure 2, our MotionFormer consists of two encoders and one decoder. The encoders extract feature pyramids from the source person image and the target pose image, respectively. These feature pyramids are sent to the decoder for accurate matching and motion transfer. In the decoder, there are several decoder blocks and one fusion block. Each block consists of two parallel branches (i.e., the warping and generation branches). In the warping branch, we predict a flow field to warp features from the source person image, which preserves the information of the source image to achieve high-fidelity motion transfer. Meanwhile, the generation branch produces novel content that cannot be directly borrowed from the source appearance to improve photorealism further. Afterward, we use a fusion block to convert the feature output of these two branches into the final transferred image.

The feature matching result dominates the flow field and generation content. Specifically, we implement feature matching from two input images via convolutions and cross-attentions in each branch. The tokens from the source image features are mapped into Key and Value, and the tokens from the target pose image features are mapped into Query. We compute the cross-attention map based on the Key and Query. This map reflects the global correlations between two input images. Then, we send the output of cross-attention process into a convolution to capture locally matched results. Thanks to

the accurate global and local feature matching, our MotionFormer can improve the performance of both the warping and generation processes. Moreover, we propose a mutual learning loss to enable these two branches to supervise each other during training, which helps them to benefit from each other and facilitates the final fusion for high-quality motion transfer. In the test phase, our method generates a motion transfer video on the fly given a single source image without training a person-specific model or fine-tuning. Some results generated by our method can be found in Fig. 1, and we show more video results in the supplementary files.

## 2 RELATED WORKS

**Human Motion Transfer.** Most human motion transfer works are built upon an image-to-image Isola et al. (2017); Wang et al. (2018b) or video-to-video Wang et al. (2018a; 2019) translation framework. With this framework, some methods Chan et al. (2019); Esser et al. (2018); Dong et al. (2018); Han et al. (2019); Ma et al. (2017); Ren et al. (2021; 2020) set the off-the-shelf 2D body keypoints Cao et al. (2017); Xiu et al. (2018) as a condition to animate the source person image. And some methods Siarohin et al. (2019a;b; 2021) use the unsupervised 2D body keypoints to extract motion representations. Therefore, these methods generalize well to a wider range of objects(i.e., animals). The 2D body keypoints can represent the body motion correctly, but this representation is sparse and the CNN models used by these methods have limited receptive fields, which leads to inaccurate global visual correspondence between this motion representation and the source image. To overcome this limitation, recent approachesNeverova et al. (2018); Grigorev et al. (2019); Huang et al. (2021a); Shysheya et al. (2019); Liu et al. (2019b); Ma et al. (2018) project the source person and the target person into a unified 3D space (i.e., DensePose Alp Güler et al. (2018) and SMPL Loper et al. (2015)) to capture pixel-level correspondences, which help render the appearance from source to target in global perspective. And they use the inpainting methodsLiu et al. (2019a; 2020; 2021a;b) to restore the background image. However, compared to 2D keypoint representations, DensePose and SMPL models are less accurate and may produce large misalignments in complex scenes. In this paper, we design the first Vision-Transformer-based generation framework for human motion transfer. With merely 2D keypoints guiding the target motion, we capture large motion deformations with globally attended appearance and motion features and yield state-of-the-art performance. Unlike many previous methods that trained on a single video Chan et al. (2019); Shysheya et al. (2019) or need fine-tuning Huang et al. (2021a); Liu et al. (2019b); Zakharov et al. (2019) to achieve higher perceptual quality, our method works in a one-shot fashion that directly generalizes to unseen identities.

**Vision Transformer.** Transformer Vaswani et al. (2017) has become increasingly popular in solving the computer vision problems, such as object detection Carion et al. (2020); Liu et al. (2021c); Touvron et al. (2021), segmentation Wang et al. (2021a); Zheng et al. (2021); Cao et al. (2021b), in-painting Peng et al. (2021); Wan et al. (2021), image generation Cao et al. (2021a); Lee et al. (2021); Jiang et al. (2021); Esser et al. (2021), restoration Chen et al. (2021b); Wang et al. (2021b); Liang et al. (2021); Zhu et al. (2022), image classification Wang et al. (2021a); Huang et al. (2021b); Ge et al. (2021); Liu et al. (2021c); Wu et al. (2021); Dong et al. (2022); Liang et al. (2022); Chen et al. (2022; 2021a), and 3D human texture estimation Xu & Loy (2021). Due to the powerful global information modeling ability, these Transformer-based methods achieve significant performance gain compared with CNNs that focus on local information. In this paper, we successfully manage to take advantages of Vision Transformers for motion transfer with a warping and generation two branch architecture. The two branches employ cross-attention and convolution to enrich generation quality from both global and local viewpoints. Besides, we propose a novel mutual learning loss to regularize two branches to learn from each other, which effectively increases photorealism.

## 3 PROPOSED METHOD

Fig. 2 shows an overview of MotionFormer. It consists of two Transformer encoders and one Transformer decoder. The two Transformer encoders first extract the features of source image $I_s$ and target pose image $P_t$, respectively. Then the Transformer decoder builds the relationship between $I_s$ and $P_t$ with two-branch decoder blocks hierarchically. Finally, a fusion block predicts the reconstructed person image $I_{out}$. The network is trained end-to-end with the proposed mutual learning loss. We utilize the Cross-Shaped Window Self-Attention (CSWin Attention) Dong et al. (2022)

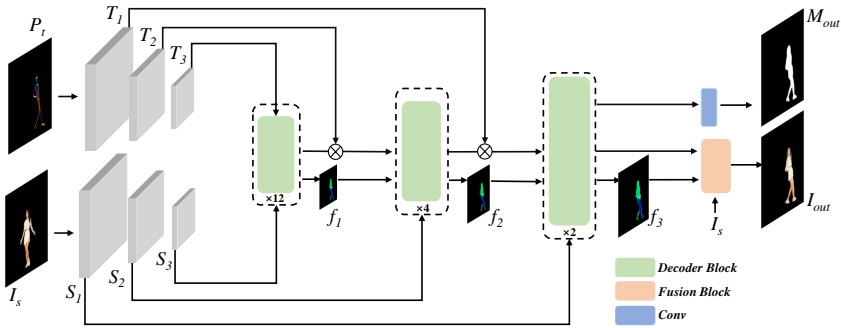

Figure 2: Overview of our MotionFormer framework. We use two Transformer encoders to extract features of the source image $I_s$ and the target pose image $P_t$. These two features are hierarchically combined in one Transformer decoder where there are multiple decoder blocks. Finally, a fusion block synthesizes the output image by blending the warped and generated images.

as our Attention mechanism in the encoder and decoder. The CSWin Attention calculates attention in the horizontal and vertical stripes in parallel to ensure the performance and efficiency. In our method, we assume a fixed background and simultaneously estimate a foreground mask $M_{out}$ to merge an inpainted background with $I_{out}$ in the testing phase. We introduce the Transformer encoder and decoder in Sec. 3.1 and Sec. 3.2 respectively. The mutual learning loss is in Sec. 3.3.

## 3.1 TRANSFORMER ENCODER

The structure of the two Transformer encoders is the same. Each encoder consists of three stages with different spatial size. Each stage has multiple encoder blocks and we adopt the CSwin Transformer Block Dong et al. (2022) as our encoder block. Our Transformer encoder captures hierarchical representations of the source image $I_s$ and the target pose image $P_t$. $S_i$ and $T_i$ ($i = 1, 2, 3$) denote the output of the $i$-th stage for $I_s$ and $P_t$, respectively, as shown in Fig. 2. We follow Dong et al. (2022); Wu et al. (2021) to utilize a convolutional layer between different stages for token reduction and channel increasing. We show more details of the encoder in the appendix.

## 3.2 TRANSFORMER DECODER

There are three stages in our Transformer decoder. The number of the decoder block in each stage is 2, 4, and 12, respectively. We concatenate the output of each stage and the corresponding target pose feature by skip-connections. The concatenated results are sent to the second and third stages. Similar to the encoder, we set the convolutional layer between different stages to increase token numbers and decrease channel dimensions.

### 3.2.1 DECODER BLOCK.

As shown in Fig 3, the decoder block has warping and generation branches. In each branch, there is a cross-attention process and a convolutional layer to capture the global and local correspondence respectively. Let $X_{de}^l$ denote the output of $l$-th decoder block ($l > 1$) or the output of precedent stage ($l = 1$). For the first decoder stage, we set the $T_3$ as input so the $X_{de}^1 = T_3$. The decoder block first extracts $\hat{X}_{de}^l$ from $X_{de}^{l-1}$ with a Multi-Head Self-Attention process. Then we feed $\hat{X}_{de}^l$ to the warping branch and generation branch as Query ($Q$), and we use the feature of source encoder $S_i$ as Key ($K$) and Value ($V$) to calculate the cross-attention map similar to Vaswani et al. (2017) with Multi-Head Cross-Attention process. The cross-attention map helps us build the global correspondence between the target pose and the source image. Finally, we send the output of the Multi-Head Cross-Attention to a convolutional layer to extract the local correspondence. The warping branch predicts a flow field to deform the source feature conditioned on the target pose, which helps the generation of regions that are visible in the source image. While for the invisible parts, the generation branch synthesizes

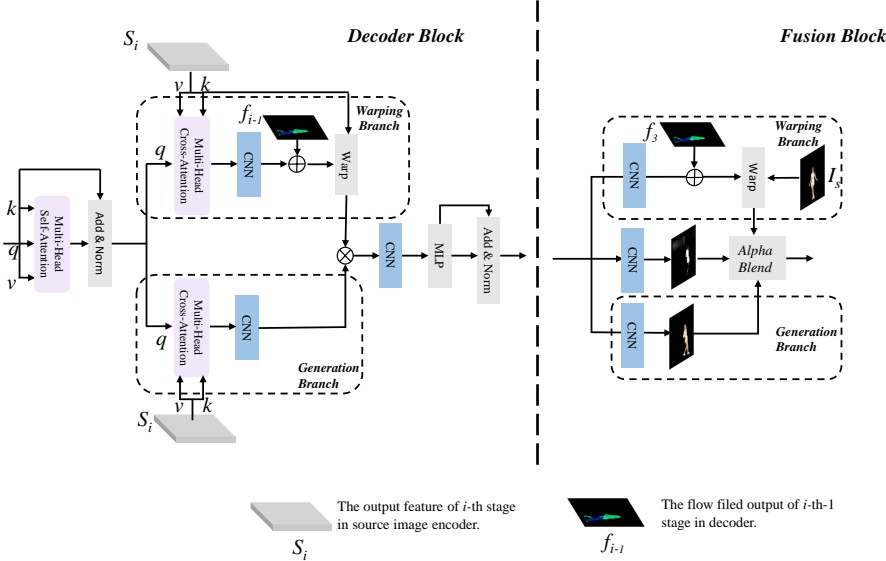

Figure 3: Overview of our decoder and fusion blocks. There are warping and generation branches in these two blocks. In decoder block, We build the global and local correspondence between source image and target pose with Multi-Head Cross-Attention and CNN respectively. The fusion block predict an mask to combine the output of two branches in pixel level.

novel content with the contextual information mined from the source feature. We combine the advantages of these two branches in each decoder block to improve the generation quality.

**Warping branch.** The warping branch aims to generate a flow field to warp the source feature $S_i$. Specifically, the Multi-Head Cross-Attention outputs the feature with the produced $Q$, $K$, $V$, and we feed the output to a convolution to inference the flow field. Inspired by recent approaches that gradually refine the estimation of optical flow Hui et al. (2018); Han et al. (2019), we estimate a residual flow to refine the estimation of the previous stage. Next, we warp the feature map $S_i$ according to the flow field using bilinear interpolation. Formally, the whole process can be formulated as follows:

$$
\begin{aligned}
Q &= W^Q(\hat{X}_{de}^l), K = W^K(S_i), V = W^V(S_i), \\
f^l &= \text{Conv}(\text{Multi-Head Cross-Attention}(Q, K, V))), \\
f^l &= \text{Up}(f_{i-1}) + f^l, \text{if } l = 1 \text{ and } i > 1, \\
O_w^l &= \text{Warp}(S_i, f^l),
\end{aligned}
\tag{1}
$$

where $W^Q, W^K, W^V$ are the learnable projection heads in the self-attention module, the $O_w^l$ denotes the output of the warping branch in $l$-th block, the Up is a $\times 2$ nearest-neighbor upsampling, and Warp denotes warping feature map $S_i$ according to flow $f^l$ using grid sampling Jaderberg et al. (2015). For the $i$-th decoder stage, the flow predicted by the last decoder block is treated as $f_i$ and then refined by the subsequent blocks.

**Generation branch.** The architecture of the generation branch is similar to the warping branch. The attention outputs the feature with the produced $Q$, $K$, $V$, and then we feed the output to a convolution to infer the final prediction $O_g^l$:

$$
\begin{aligned}
Q &= W^Q(\hat{X}_{de}^l), K = W^K(S_i), V = W^V(S_i), \\
O_g^l &= \text{Conv}(\text{Multi-Head Cross-Attention}(Q, K, V))).
\end{aligned}
\tag{2}
$$

where $W^Q, W^K, W^V$ are the learnable projection heads in the self-attention module. The generation branch can generate novel content based on the global information of source feature $S_i$. Therefore, it is complementary to the warping branch when the flow field is inaccurate or there is no explicit reference in the source feature. Finally, we concatenate the output of warping and generation branch and reduce the dimension with a convolutional layer followed by an MLP and a residual connection:

$$\bar{X}_{de}^l = \text{Conv}(\text{ Concat }(O_w^l, O_g^l)),$$
$$X_{de}^l = \text{MLP}(\text{LN}(\hat{X}_{de}^l)) + \bar{X}_{de}^l, \tag{3}$$

where the $\bar{X}_{de}^l$ is the combination of warping and generation branches in the $l$-th decoder block.

### 3.2.2 FUSION BLOCK.

The fusion block takes the decoder output to predict the final result. The fusion block has a warping branch and generation branch at the pixel level. The warping branch refines the last decoder flow field $f_3$ and estimates a final flow $f_f$. And the generation branch synthesizes the RGB value $I_f$. At the same time, a fusion mask $M_f$ is predicted to merge the output of these two branches:

$$f_f = \text{Conv}(O_{de}) + \text{Up}(f_3),$$
$$M_f = \text{Sigmoid}(\text{Conv}(O_{de})),$$
$$I_f = \text{Tanh}(\text{Conv}(O_{de})), \tag{4}$$
$$I_{out} = M_f \odot \text{Warp}(I_s, f_f) + (1 - M_f) \odot I_f,$$

where the $O_{de}$ is the output of decoder, $\odot$ is the element-wise multiplication, and $I_{out}$ is the final prediction.

### 3.3 MUTUAL LEARNING LOSS

The generation and warping branches have their own advantages as mentioned above. Intuitively, we concatenate the output of these two branches followed by a convolution layer and an MLP as shown in Fig. 2, but we empirically find the convolution layer and MLP cannot combine these advantages well (see Sec. 5). To address this limitation and ensure the results have both advantages of these two branches, we propose a novel mutual learning loss to enforce these two branches to learn the advantages of each other. Specifically, the mutual learning loss enables these two branches to supervise each other within each decoder block, let $O_w^k, O_g^k \in R^{(H \times W) \times C}$ denote the reshaped outputs of the last warping and generation branch at the $k$-th decoder stage (see Eqs. (1) and (2) for their definition). If we calculate the similarity between the feature vector $O_{w,i}^k \in R^C$ at the spatial location $i$ of $O_w^k$ and all feature vectors $O_{g,j}^k \in R^C$ ($j = 1, 2, \ldots, HW$) in $O_g^k$, we argue that the most similar vector to $O_{w,i}^k$ should be $O_{g,i}^k$, which is at the same position in $O_g^k$. In another word, we would like to enforce $i = \arg\max_j \text{Cos}(O_{w,i}^k, O_{g,j}^k)$, where $\text{Cos}(\cdot, \cdot)$ is the cosine similarity. This is achieved by the following mutual learning loss:

$$L_{\text{mut}} = \sum_k \sum_{i=1}^{HW} ||\text{SoftArgMax}_j(\text{Cos}(O_{w,i}^k, O_{g,j}^k)) - i||_1, \tag{5}$$

where the SoftArgMax is a differentiable version of $\arg\max$ that returns the spatial location of the maximum value. The mutual learning loss constrain the two branches to have high correlations at the same location, enhancing the complementariness of warping and generation.

In addition to the perceptual diversity loss, we follow the Ren et al. (2020) and Huang et al. (2021a) utilize the reconstruction loss Johnson et al. (2016), feature matching loss Wang et al. (2018c), hinge adversarial loss Lim & Ye (2017), style loss Gatys et al. (2015), total variation loss Johnson et al. (2016) and mask loss Huang et al. (2021a) to optimize our network. Details are in appendix.

## 4 EXPERIMENTS

**Datasets.** We use the solo dance YouTube videos collected by Huang et al. (2021a) and iPer Liu et al. (2019b) datasets. These videos contain nearly static backgrounds and subjects that vary in gender, body shape, hairstyle, and clothes. All the frames are center cropped and resized to $256 \times 256$. We train a separate model on each dataset to fairly compare with other methods.

**Implementation details.** We use OpenPose Cao et al. (2017) to detect 25 body joints for each frame. These joints are then connected to create a target pose stick image $P_t$, which has 26 channels and

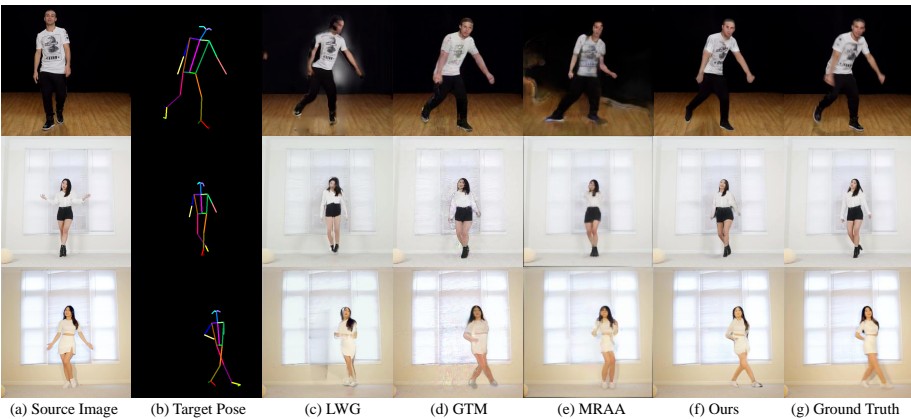

(a) Source Image    (b) Target Pose    (c) LWG    (d) GTM    (e) MRAA    (f) Ours    (g) Ground Truth

Figure 4: Visual comparison of state-of-the-art approaches and our method on YouTube videos dataset. Our proposed framework generates images with the highest visual quality.

each channel indicating one stick of the body. We use $M_{gt}$ to separate foreground person image $I_{gt}$ from the original frame. Our model is optimized using Adam optimizer with $\beta_1 = 0.0$, $\beta_2 = 0.99$, and initial learning rate of $10^{-4}$. We utilize the TTUR strategy Heusel et al. (2017) to train our model. During the inference phase, we select one image from one video as the source image. The target background is generated by inpainting the source image background with LAMA Suvorov et al. (2021). When the target person has different body shapes (*e.g.*, heights, limb lengths) or the target person and source person are at different distances from the camera, we use the strategy in Chan et al. (2019) to normalize the pose of the target human.

**Baselines.** We compare MotionFormer with the state-of-the-art human motion transfer approaches: LWG Liu et al. (2019b), GTM Huang et al. (2021a), MRAA Siarohin et al. (2021) and DIST Ren et al. (2020). For LWG, we test it on iPer dataset with the released pre-trained model, and we train LWG on YouTube videos with its source code. At test time, we fine-tune LWG on the source image as official implementation (fine-tuning is called "personalize" in the source code). For GTM, we utilize the pre-trained model on the YouTube videos dataset provided by the authors and retrain the model on iPer with the source code. As GTM supports testing with multiple source images, we use 20 frames in the source video and fine-tune the pre-trained network as described in the original paper Huang et al. (2021a). For MRAA, we use the source code provided by the authors to train the model. For DIST Ren et al. (2020), we compare with it using the pre-trained model on iPer dataset. For synthesizing a 1,000 frame video, the average per frame time costs of MRAA, LWG, GTM, DIST, and our method are 0.021s, 1.242s, 1.773s, 0.088s, and 0.94s, respectively. Meanwhile, MotionFormer does not require an online fine-tuning, while LWG and GTM do.

## 4.1 QUALITATIVE COMPARISONS

Qualitatively comparisons are given in Fig. 4 and Fig. 5. Although LWG Liu et al. (2019b) can maintain the overall shape of the human body, it fails to reconstruct complicated human parts (*e.g.*, long hair and shoes in Fig. 4) of the source person and synthesis image with a large body motion (*e.g.*, squat in the red box of Fig. 4), which leads to visual artifacts and missing details. This is because LWG relies on the 3D mesh predicted by HMR Kanazawa et al. (2018), which is unable to model detailed shape information. In contrast, GTM Huang et al. (2021a) reconstructs better the body shape as it uses multiple inputs to optimize personalized geometry and texture. However, the geometry cannot handle the correspondence between the source image and the target pose. The synthesized texture also presents severe artifacts, especially for invisible regions in the source images. As an unsupervised method, MRAA Siarohin et al. (2021) implicitly models the relationship between source and target images. Without any prior information about the human body, MRAA generates unrealistic human images. DIST Huang et al. (2021a) does not model correct visual correspondence (*e.g.*, the coat buttons are missing in the last example) and suffers from overfitting (*e.g.*, the coat color becomes dark blue in the third example). Compared to existing methods, MotionFormer renders more realistic and natural images by effectively modeling long-range correspondence and local details.

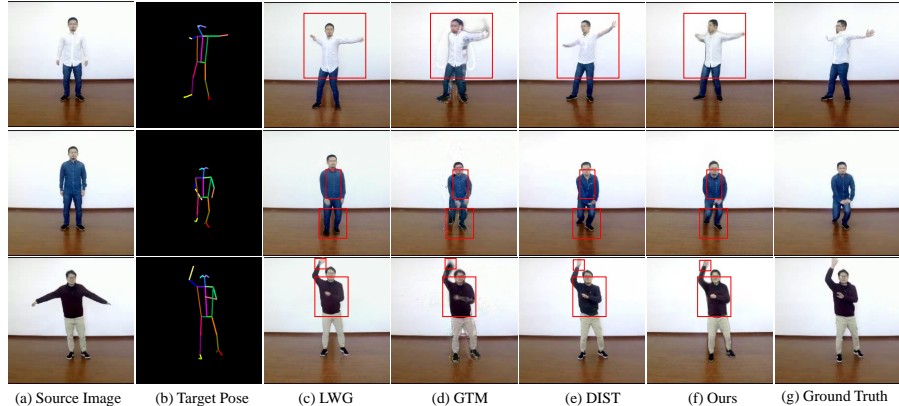

| (a) Source Image | (b) Target Pose | (c) LWG | (d) GTM | (e) DIST | (f) Ours | (g) Ground Truth |

Figure 5: Visual comparison of state-of-the-art approaches and our method on iPer dataset. Our proposed framework generates images with the highest visual quality.

| Method | PSNR↑ | SSIM↑ | LPIPS↓ | FID↓ | User study↓ |
|---|---|---|---|---|---|
| LWG Liu et al. (2019b) | 18.94 | 0.686 | 0.175 | 85.06 | 97.61% |
| GTM Huang et al. (2021a) | 21.50 | 0.819 | 0.137 | 77.69 | 76.19% |
| MRAA Siarohin et al. (2021) | 18.95 | 0.674 | 0.234 | 160.97 | 100% |
| Ours | **23.50** | **0.885** | **0.073** | **65.03** | - |

Table 1: Quantitative comparisons of state-of-the-art methods on YouTube videos dataset. User study denotes the preference rate of our method against the competing methods. Chance is 50%.

## 4.2 QUANTITATIVE COMPARISONS

We use SSIM Wang et al. (2004), PSNR, FID Heusel et al. (2017), and LPIPS Zhang et al. (2018) as numerical evaluation metrics. The quantitative results are reported in Table 1 and Table 2. We observe that our method outperforms existing methods by large margins across all metrics. Additionally, we perform a human subjective evaluation. We generate the motion transfer videos of these different methods by randomly selecting 3-second video clips in the test set. On each trial, a volunteer is given compared results on the same video clip and is then asked to select the one with the highest generation quality. We tally the votes and show the statistics in the last column of Table 1 and Table 2. We can find that our method is favored in most of the trials.

## 5 ABLATION STUDY

**Attention Mechanism.** To evaluate the effects of the cross-attention module, we delete the cross-attention in both the warping and generation branch. Instead, we concatenate the source feature $S_i$ and Query directly in the Transformer decoder, followed by a convolution layer constructing their local relationship (this experiment is named Ours w/o Attention). As shown in Fig. 6(c), without modeling the long-range relationship between the source and target, Ours w/o Attention achieves worse results (*e.g.*, distorted skirt, limbs, and shoes). The numerical comparison shown in Table 3 is consistent with the visual observation.

**Generation and warping branches.** We show the contributions of the generation branch and warping branch in the decoder block by removing them individually (*i.e.*, Ours w/o warping, Ours w/o generation). As shown in Fig. 6(d), without the warping branch, the generated clothing contains unnatural green and black regions in the man's T-shirt and woman's skirt, respectively. This phenomenon reflects that a single generation branch is prone to over-fitting. On the other hand, the warping branch can avoid over-fitting as shown in Fig. 6(e). However, the results still lack realism as the warping branch cannot generate novel appearances which are invisible in the source image (*e.g.*, the shoes of the man and the hair of the woman are incomplete). Our full model combines

| Method | PSNR↑ | SSIM↑ | LPIPS↓ | FID↓ | User study↓ |
|---|---|---|---|---|---|
| LWG Liu et al. (2019b) | 23.93 | 0.843 | 0.089 | 40.41 | 58.90% |
| GTM Huang et al. (2021a) | 23.94 | 0.840 | 0.120 | 60.26 | 75.00% |
| DIST Ren et al. (2020) | 24.19 | 0.852 | 0.071 | 30.34 | 62.50% |
| Ours | **24.72** | **0.856** | **0.069** | **27.25** | - |

Table 2: Quantitative comparisons of state-of-the-art methods on iPer videos dataset. User study denotes the preference rate of our method against the competing methods. Chance is 50%.

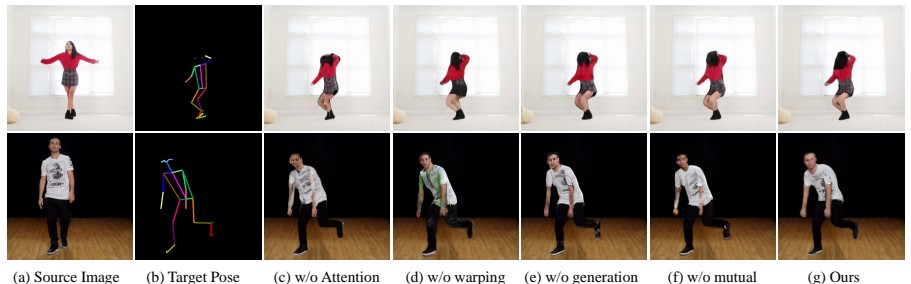

(a) Source Image    (b) Target Pose    (c) w/o Attention    (d) w/o warping    (e) w/o generation    (f) w/o mutual    (g) Ours

Figure 6: Visual ablation study on YouTube videos dataset. (a) The source image. (b) The target pose. (c) Our method without Attention. (d) Our method without the warping branch. (e) Our method without the generation branch. (f) Our method without the mutual learning loss. (g) Our full method. Our full model can generate realistic appearance and correct body pose.

the advantages of these two branches and produces better results in Fig. 6(g). We also report the numerical results in Table 3, in which our full method achieves the best performance.

**Mutual learning loss.** We analyze the importance of mutual learning loss (Eq. (5)) by removing it during training (Ours w/o mutual). Fig. 6(f) shows the prediction combining the advantages of both warping and generation branches without using the mutual learning loss, which still produces noticeable visual artifacts. The proposed mutual learning loss aligns the output features from these two branches and improves the performance. The numerical evaluation in Table 3 also indicates that mutual learning loss improves the generated image quality. The other loss terms have been demonstrated effective in Balakrishnan et al. (2018); Wang et al. (2018b); Liu et al. (2019b) with sufficient studies, so we do not include them in the ablation studies.

| Method | PSNR↑ | SSIM↑ | LPIPS↓ | FID↓ |
|---|---|---|---|---|
| Ours w/o attention | 20.95 | 0.842 | 0.194 | 81.10 |
| Ours w/o warping | 22.81 | 0.873 | 0.083 | 73.11 |
| Ours w/o generation | 22.39 | 0.872 | 0.100 | 70.21 |
| Ours w/o mutual | 22.73 | 0.876 | 0.078 | 68.70 |
| Ours | **23.50** | **0.885** | **0.073** | **65.03** |

Table 3: Ablation analysis of our proposed method on YouTube dataset. Our Full method achieves results that are superior to all other variants.

## 6    CONCLUDING REMARKS

In this paper, we introduce MotionFormer, a Transformers-based framework for realistic human motion transfer. MotionFormer captures the global and local relationship between the source appearance and target pose with carefully designed Transformer-based decoder blocks, synthesizing promising results. At the core of each block lies a warping branch to deform the source feature and a generation branch to synthesize novel content. By minimizing a mutual learning loss, these two branches supervise each other to learn better representations and improve generation quality. Experiments on a dancing video dataset verify the effectiveness of MotionFormer.

ETHIC DISCUSSIONS

This work introduces a motion transfer method that can transfer motions from one subject to another. It may raise the potential ethical concern that malicious actions can be transferred to anyone (*e.g.*, celebrities). To prevent action retargeting on celebrities, we may insert a watermark to the human motion videos. The watermark contains the original motion source, which may differentiate the celebrity movement. Meanwhile, we can construct a celebrity set. We first conduct face recognition on the source person; if that person falls into this set, we will not perform human motion transfer.

REPRODUCIBILITY STATEMENT

The MotionFormer is trained for 10 epochs, and the learning rate decays linearly after the 5-th epoch. We provide the pseudo-code of the training process in Algorithm 1. We denote the Transformer encoder of the source images as $En_s$, the Transformer encoder of the target pose as $En_t$, the Transformer decoder as $De$, and the discriminator as $D$. We set the batchsize as 4 and $step_{max}$ is obtained by dividing the image numbers of dataset by batchsize. Meanwhile, we show the details of the model architecture and loss function in the appendix, this information is useful for the reproduce process.

---

**Algorithm 1** Training Process

---

**Require:** A set of source images $I_s$, target pose images $P_t$, and person mask images $M_{gt}$.
  **for** Epoch = 1, 2, 3, ..., 10 **do**
    **for** Step = 1, 2, 3, ..., $step_{max}$ **do**
      Sample a batch of source images $I_s$, target pose $P_t$, and person mask $M_{gt}$.
      Get $S_1, S_2, S_3 = En_s(I_s), T_1, T_2, T_3 = En_t(P_t)$;
      Get $I_{out}, M_{out}, f_f = De([S_1, S_2, S_3], [T_1, T_2, T_3])$;
      Calculate the adversarial loss in Equation (6) in the appendix;
      Update $D$;
      Calculate the loss in Equation (9) in the appendix;
      Update $De$, $En_s$, $En_t$;
    **end for**
    **if** Epoch $\geq$ 5 **then**
      Update learning rate;
    **end if**
  **end for**

---

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
