# OpenReview forum: "Human MotionFormer: Transferring Human Motions with Vision Transformers"
_ICLR.cc/2023/Conference — ICLR 2023 poster_

### Official Review · Reviewer_8CDv · 2022-10-19

**Confidence:** 4
**Correctness:** 3
**Technical Novelty And Significance:** 3
**Empirical Novelty And Significance:** 2
**Recommendation:** 8

**Clarity, Quality, Novelty And Reproducibility:**

The writing is clear, technical quality is good, the proposed method is somewhat novel and the code is provided as well.

**Strength And Weaknesses:**

+ Global and local perception design within both warping and generation branches for high quality motion retargeting synthesis. A combination of global perception via cross-attentions (i.e., vision transformers) and local perception via conv layer (i.e., CNN) seems intuitive and demonstrated effective in practice.
+ The design of mutual learning loss combines the advantage of cross-attention and convolutions to enhance the warping and generation completeness. This is illustrated important to enable two branches work with each other to produce correlated visual contents.
+ High quality results shown in the experiments and supplement video.


**Summary Of The Paper:**

A motion retargeting method based on vision transformer is proposed in this paper. This task is formulated into the pattern matching problem where global and local search are required in one human image. Considering the local perception of conv layer and global perception of cross attentions, this paper combines the advantage of both. The global and local perception are leveraged into both warping and generation branches of the decoder. This design brings higher quality warping and generation results, and finally produces higher quality motion retargeted human images.

**Summary Of The Review:**

Overall, this is a nice framework combining vision transformers and conv layers for global and local matching to synthesize high quality motion retargeting results. Experiments have shown the effectiveness of the proposed method upon prior arts.

---

> ### Author Response · Authors · 2022-11-17
> **Response to Reviewer 8CDv**
>
> Thank you very much for your appreciation of our paper. We want to bring a new perspective to the motion transfer task with vision transformers, and we believe the MotionFormer can open up the minds of other researchers.

---

### Official Review · Reviewer_6Q61 · 2022-10-25

**Confidence:** 5
**Correctness:** 3
**Technical Novelty And Significance:** 2
**Empirical Novelty And Significance:** 2
**Recommendation:** 3

**Clarity, Quality, Novelty And Reproducibility:**

The technical writing and presentation need significant improvement. Multiple grammar and formatting mistakes can be found in the manuscript. Besides, the presentation falls into trivial details and lacks an in-depth discussion of the problem.

**Details Of Ethics Concerns:**

The authors claim that "professional actions can be transferred to celebrities", which surely brings about ethical concerns. However, no related discussions are found.

**Strength And Weaknesses:**

Strength: The two branches (generation+warping) seem reasonable and could benefit each other (ablation results).

Weaknesses:
1. The writing of this paper could be improved. The abstract and introduction section fails to address key challenges in the field and fall into trivial details.
2. The technical novelty is limited. The major contribution lies in the transformer decoder blocks (Sec. 3.2), and the techniques are not very novel in this task.
3. Lack of clarification. The target pose is used to guide the source image, but no discussion is provided on how the target pose is extracted, represented, and how its quality affects the final results. What if the target pose and source image have large structure variations? Is there a mechanism like [Chan et al. (2019);] to align them?




**Summary Of The Paper:**

This paper proposes Human MotionFormer, which utilizes convolutional layers and cross attentions for local and global matching. The network could achieve both warping and generation, posing the source image to the target pose.

**Summary Of The Review:**

This paper is not ready for publication at ICLR at this stage due to limited novelty and writing quality.

---

> ### Author Response · Authors · 2022-11-17
> **Response to Reviewer 6Q61**
>
> ***1. Clarity.***
>
> We have significantly refined our abstract, introduction, and
> related work sections to elucidate our observations, motivations, and objectives. The remaining part of the paper is also proofread to fix unclear presentations. Our modifications are marked in red.
>
> ***2. Novelty.***
>
> We have noticed that decomposing motion transfer as warping and generation has been investigated previously [1, 2, 3].
> The main difference between ours and existing studies is that our MotionFormer focus on designing accurate global and local matchings simultaneously between the target pose and the source person, which is not well addressed by existing methods.
>
> These two matchings are important because they are the foundation of motion transfer. To the best of our knowledge, we are the $\textbf{\textit{first}}$ to introduce hierarchical vision transformers (ViTs) into human motion transfer with the insights from ViTs. Our MotionFormer uses the encoder of ViT to capture the feature of the target pose (i.e.,  $P_t$) and source image (i.e., $I_s$). Then our decoder block is used to model the global and local matching between $P_t$ and $I_s$. A cross-attention mechanism models the global matching, and the local matching is perceived by convolution. Moreover, the hierarchical structure can gradually refine matching accuracy to improve the output's quality of warping and generation branches. Finally, a fusion module combines the final outputs of warping and generation branches to synthesize the motion transfer results.
>
> The warping and generation branches construct matching from different aspects. For the warping branch, the model builds the matching explicitly with the flow field and preserves the source image's detail. For the generation branch, the model builds the matching implicitly to produce novel content when the warping branch cannot borrow information from the source appearance. We observe that directly combining the outputs of these two branches cannot improve their strengths simultaneously since they synthesize the image from different perspectives (see w/o mutual in the ablation study). So we propose a $\textbf{mutual learning loss}$ to enable these two branches to supervise each other during training, which helps them to benefit from each other and facilitates the final fusion for high-quality motion transfer. Moreover, the warping and generation modules are either serial in the feature space [1, 2] or parallel in the image space [3]. Our decoder and fusion modules perform warping and generation in parallel in both spaces, bringing more complete perceptions of images and semantics. Quantitative and qualitative results indicate the superior performance of our method. The supplementary video can also prove that our method has a good effect.
>
> [1] Liu, Wen, et al. "Liquid warping gan: A unified framework for human motion imitation, appearance transfer and novel view synthesis." In Proceedings of the IEEE/CVF International Conference on Computer Vision. 2019.
>
> [2] Siarohin, Aliaksandr, et al. "Motion representations for articulated animation." In Proceedings of the IEEE/CVF Conference on Computer Vision and Pattern Recognition. 2021.
>
> [3] Ren, Yurui, et al. "Combining attention with flow for person image synthesis." In Proceedings of the 29th ACM International Conference on Multimedia. 2021.
>
>
>
> ***3. Clarification.***
>
> As mentioned in the "Implementation details" paragraph, we use OpenPose to detect 25 body joints for each frame. These joints are then connected to create a target pose stick image $P_t$, which consists of 26 channels where each channel indicates one stick of the body.
>
> If the target pose is incorrect, the synthesized human may contain inaccurate actions in the final results. We will use the more robust pose detection method (Alphapose [1]) in the future.
>
>
> We guess what you refer [Chan et al. (2019);] means [2]. We apologize for not describing the situation's details when the target human has a different limb proportion from the source human. In this situation, we follow a mechanism similar to [2] for alignment.
>
> [1] Xiu, Yuliang, et al. "Pose Flow: Efficient online pose tracking." arXiv preprint arXiv:1802.00977 (2018).
>
> [2] Chan, Caroline, et al. "Everybody dance now." Proceedings of the IEEE/CVF international conference on computer vision. 2019.
>
>
> ***4. Ethical Concerns.***
>
> Thank you for your suggestion, and we have added an "Ethical Discussion' section to our paper.

---

> > ### Comment · Reviewer_6Q61 · 2022-11-21
> > **Response to authors**
> >
> > Thank the authors for the detailed reply and the updates.
> >
> > My concerns are still about the contribution of MotionFormer.
> > 1. Previous work DIST seems to have better visualization quality than the proposed method, especially in keeping the identity of the person, and the reflection of the targeted pose of the human. See Figure 5 & 11 and the supplementary video. I think the reflection failure may be caused by human mask loss.
> > 2. The warping and generation scheme is not novel, and the warping design in MotionFormer still fails to preserve the source image details (the human identity is not the same in the face and the folds of the clothes also disappear), which is not convincing.
> >
> > Other thoughts
> > The low visual quality may be due to the losing depth information of the pose image $P_t$, i.e. which body joint is closer to the camera. Therefore, I'm doubting whether it is the right way to only use $P_t$ as guidance.
> >
> > In summary, I think this paper could still be polished in the design for better generation results both in identities and high-fidelity details and is not ready for publication at this stage due to limited novelty.

---

> > > ### Author Response · Authors · 2022-11-21
> > > **Response to Reviewer 6Q61**
> > >
> > >
> > > Dear reviewer,
> > >
> > > Thank you for the further comments and suggestions on our paper. We clarify your concerns below:
> > >
> > > - We would like to respectfully disagree with your concerns about the contribution and novelty of our paper. First, DIST suffers from identity overfitting, as shown in Figure 5(i.e., the coat color becomes dark blue in the third example) and Figure 11(i.e., the shoes become red in the second example). Thus, it is not "keeping the identity of the person," but it fails to generalize a new identity given the source person image. The extensive comparison with a user study in Table 2 further validates that our method outperforms DIST. Meanwhile, as mentioned in our paper, motion transfer's main challenge is capturing the global and local matching between the source image and target pose, which is essential to make the source person do the correct target motion. However, DIST does not capture the correct matching between the source image and target pose (i.e., the arm in both the last example in  Figure 11  and the first example in  Figure 5), and our MotionFormer performance is better. Second, we have already made it crystal clear that the warping and generation scheme is not our main contribution. Suppose you insist that any paper that uses a warping and generation scheme for human motion transfer is not novel. In that case, we doubt that most of the former papers should be considered non-novel, and there will not be any more novel papers for this task. We encourage you to focus on the real merits of our paper instead of a commonly adopted strategy of most related papers.
> > >
> > > - About the reflection failure, we agree that it is due to our scheme of generating the foreground and background seperately with a human mask loss. But this is just a design choice following [1] and not the main focus of our paper. Furthermore, our method could be readily extended to work on foreground and background generation to fix such issues. We will demonstrate the results of generating non-human regions with MotionFormer in a revised version. We believe the ability to model global and local attention of MotionFormer will also benefit the background generation.
> > >
> > > - Indeed, using depth information that cannot be modeled by the pose image $P_t$ will reduce the depth ambiguity and lead to better performance. However, it also makes the comparison unfair as most methods merely use this 2D guidance for image generation. We will illustrate the results using 3D keypoints in the final version and make a comprehensive comparison with other methods. We thank you for pointing this out.
> > >
> > > - In sum, human motion transfer is an interesting and challenging problem in which different methods may introduce different artifacts. And MotionFormer is not an exception---it cannot avoid all artifacts and generate perfect results. However, as the first approach that makes Vision Transformer work on human motion transfer, MotionFormer successfully models the global and local correspondences with a novelly designed mutual learning loss. We believe it could receive a lot of adoption in the community and help derive new research directions.
> > >
> > > [1] Huang, Zhichao, et al. "Few-Shot Human Motion Transfer by Personalized Geometry and Texture Modeling." Proceedings of the IEEE/CVF Conference on Computer Vision and Pattern Recognition. 2021.

---

> ### Author Response · Authors · 2022-11-19
> **Sincerely Look Forward to Your Feedback**
>
> Dear Reviewer:
>
> Thanks again for all of your valuable comments and suggestions, which have helped us improve the quality and clarity of this paper!
>
>
> Since the deadline for discussion is approaching. Please let us know if you have any other doubts about our paper, we will try our best to solve your questions. We appreciate your suggestions.
>
>
> Best wishes,
>
> Authors

---

### Official Review · Reviewer_Epkc · 2022-11-02

**Confidence:** 3
**Correctness:** 4
**Technical Novelty And Significance:** 3
**Empirical Novelty And Significance:** 2
**Recommendation:** 6

**Clarity, Quality, Novelty And Reproducibility:**

Aside from the poor text quality in some parts of the paper, the methodology is clear and straightforward and the experimental results support the claims made by the authors. Moreover, the model appears to be reproducible, if the authors provide some more details of the discriminator network and the optimisation algorithm.

**Strength And Weaknesses:**

Strengths: The methodology is sound and the design choice of Vision Transformers is an interesting approach to the problem. The experiments, comparisons with SOTA and ablation studies are comprehensive and indicate a noteworthy improvement compared to previous works.

Weaknesses: The most profound shortcoming of the paper is the quality of the text. In my opinion, the Abstract, Introduction and Related Work sections require a significant revision, with the assistance of a native speaker if possible. I would strongly advise the authors to revisit these sections and improve their text. In addition, I would recommend trimming down the Abstract, as it is very long.

**Summary Of The Paper:**

This paper introduces a novel approach to the problem of human pose transfer, based on Vision Transformers (ViT). The authors propose a system with two encoding and one decoding Vision Transformers. The decoder consists of two branches, a warping branch that predicts the optical flow for warping the source image to the target pose, as well as a generation branch. Both branches possess a cross-attention module which has been shown to increase the quality of results. A mutual learning loss forces the two branches to be consistent with each other, in terms of intermediate feature map prediction. At the back end of the network, a fusion block combines the warped and generated images to form the final output. Finally, the authors provide both qualitative and quantitative comparisons against SOTA methods, along with an ablation study, which demonstrate the merits of their methodology.

**Summary Of The Review:**

The paper presents an interesting method for human pose transfer with Vision Transformers and shows promising results. My main concern has to do with the quality of the text, especially in the first two sections. Considering the above, I would suggest to accept the paper with some reservations, as the authors should revisit their text.

---

> ### Author Response · Authors · 2022-11-17
> **Response to Reviewer Epkc**
>
> ***1. Clarity.***
>
> We apologize for the confusion brought by our presentation. We have made significant changes in our abstract, introduction, and related work sections to elucidate our observations, motivations, and objectives. Meanwhile, we have thoroughly proofread our manuscript and fixed other unclear presentations and grammatical errors. Our modifications are marked in red.
>
> ***2. Reproducibility.***
>
> As mentioned in the "Implementation details" paragraph, we use the Adam optimizer with $\beta_1 = 0.0$, $\beta_2  = 0.99$, and an initial learning rate of $10^{-4}$. Meanwhile, we describe the details of the encoder, decoder and discriminator in the "Model Architecture" section of the appendix.

---

> ### Author Response · Authors · 2022-11-19
> **Sincerely Look Forward to Your Feedback**
>
> Dear Reviewer:
>
> Thanks again for all of your valuable comments and suggestions, which have helped us improve the quality and clarity of this paper!
>
>
> Since the deadline for discussion is approaching. Please let us know if you have any other doubts about our paper, we will try our best to solve your questions. We appreciate your suggestions.
>
>
> Best wishes,
>
> Authors

---

### Official Review · Reviewer_m6aS · 2022-11-06

**Confidence:** 5
**Correctness:** 4
**Technical Novelty And Significance:** 2
**Empirical Novelty And Significance:** 3
**Recommendation:** 6

**Clarity, Quality, Novelty And Reproducibility:**

The paper is most clear.  But, most of the parameters used in the paper are not instantiated. The paper does not give procedure to train the network. This makes it hard to replicate the work. Most of the component modules are well known. However, the combination of these components for the application of egocentric pose estimation is new. The overall approach is interesting.

**Strength And Weaknesses:**

Strength:
   + The proposed method uses spatial and temporal information to improve the results. The Transformer is a natural choice for such a scheme.
   + Different tricks may contribute to the overall improvement: the sequence input, the transformer, the learnable space encoding, the learnable token.

Weakness:
   -  The paper never talks about how long the input sequence needs to be. The longer should be better but it also introduces long delay, which makes the proposed method not useful for real-time applications.
  -   The writing is mostly clear. However, the details of the network are not completely included. This makes it hard to replicate the work. Most of the parameters used in the paper are not instantiated.
 -   The paper does not give procedure to train the network.

**Summary Of The Paper:**

This paper proposes a spatial temporal transformer deep network to tackle the egocentric pose estimation problem. The motivation of such a scheme is that it can help improve pose estimation when there are occlusion and large distortion.  When there is occlusion at a moment, hopefully the body part is visible at another time. The temporal pattern also helps disambiguate body poses.  The proposed method uses a resnet to extract local features. Combined with a learnable token map, they are sent to a transformer to aggregate space and time information. The transformer also uses a learnable spatial encoding. The output of the transformer than passes through a deconv layer to reconstruct a body joint heatmap. A fully connected network is then used to estimate the final 3D body pose.  Experiments show the proposed method give better results than different competing methods.

**Summary Of The Review:**

This paper proposes a transformer based method which greatly improves the egopose estimation result. The proposed method is interesting. One concern is how this method can be used for real time applications which requires a short delay if a long input sequence is needed to estimate pose at each time instant.

---

> ### Author Response · Authors · 2022-11-17
> **Response to Reviewer m6aS**
>
> We think reviewer m6aS accidentally put the review of another paper here (it seems to be a paper talking about egocentric pose estimation with spatial-temporal transformers). So, we are afraid we cannot respond properly to the concerns raised in the review. However, as human motion transfer is also a task for which it is desired to generate spatially and temporally realistic and coherent results, we would like to emphasize the following  points that reviewer m6aS may be interested in:
>
>
>
> $\bullet$ Our method can be readily extended to work in a video2video (Wang et al. 2018a) setting by also warping and compositing the previously generated frames to generate the current frame with a temporal decoder, thus leveraging rich temporal information for improved generation results.
>
> $\bullet$ By only relying on previously generated frames, the video2video version of our method can be used for real-time applications without delay.
>
> $\bullet$ The clarity and reproducibility will be improved as mentioned in the general response.

---

> ### Author Response · Authors · 2022-11-19
> **Sincerely Look Forward to Your Feedback**
>
> Dear Reviewer:
>
> Thanks again for all of your valuable comments and suggestions, which have helped us improve the quality and clarity of this paper!
>
>
> Since the deadline for discussion is approaching. Please let us know if you have any other doubts about our paper, we will try our best to solve your questions. We appreciate your suggestions.
>
>
> Best wishes,
>
> Authors

---

### Author Response · Authors · 2022-11-17
**General Reponse**


We sincerely appreciate all reviewers' efforts in reviewing our paper and giving insightful comments and valuable suggestions. We are glad that the reviewers generally acknowledge the following novelty and contributions of our work.

* ***Transformer:*** In contrast to existing works, our MotionFormer combines the advantages of both vision transformer and convolution to capture the global and local matching between the target pose and source human. This intuitive idea helps us to build robust matching and boost the performance of warping and generation to synthesize high-quality results[8CDv, Epkc].
* ***Mutual Learning Loss:*** The proposed mutual learning loss benefits the generation and warping branches to learn the advantages from each other. This helps our method to fully utilize each branch for performance improvement [8CDv, Epkc].
* ***Experiments:*** We evaluate our method in two datasets, the qualitative and quantitative analysis validated the effectiveness of our method. Meanwhile, the supplementary video further shows our appealing performance [8CDv, Epkc].





As suggested by the reviewers, we have included the following contents in our revised manuscript to improve our paper. Our modification on the paper has been marked in pink. We summarize our revisions as follows. Our detailed responses can be found in the following response sections to the reviewers.


* ***Clarity:*** We polish our abstract, introduction, and related work sections to express our contribution and motivation more clearly [8CDv, 6Q61, Epkc].
* ***Reproducibility:*** We describe the details of the encoder, decoder, and discriminator in the "Model Architecture" section of the appendix. Meanwhile, more details can be found in the "Implementation detail" of the main paper, "Reproducibility Statement", "Loss functions" and "Details of Attention Process and Encoder" in our appendix. We will release our code for public usage. [Epkc]

---

### Decision · Program_Chairs · 2023-01-20

**Decision:**

Accept: poster

**Justification For Why Not Higher Score:**

Paper with a wide spread of the scores and a strongly negative review.

**Justification For Why Not Lower Score:**

N/A

**Metareview: Summary, Strengths And Weaknesses:**

This paper proposes a simple but effective transformer-based model for motion synthesis, i.e. transferring motions from a target dynamic person to a source static person. The ViT based architecture leverages cross-attention in combination with convolution layers to build the correct global and local correspondence between the source person and a given pose of the target person.

While acknowledging that the proposed model is interesting, intuitive, and effective in practice, and the experimental evaluations are comprehensive, the reviewers have raised concerns related to 1) presentation clarity (Reviewer Epkc), 2) limited technical novelty (Reviewer 6Q61), and 3) imperfections due to reflection (Reviewer 6Q61). (1) and (2) have been improved and clarified to some extend in the rebuttal. Among these, (3) did not have a substantial impact on the decision, but would be helpful to address in a subsequent revision.
The review posted by mistake by Reviewer m6aS was not taken into account in the final recommendation.

After further discussions with Senior AC, our final recommendation is acceptance. We urge the authors to further improve presentation clarity, i.e. to include clarifications of the rebuttal into the manuscript, highlighting the responses to the main concerns in the main paper.


**Note From Pc:**

if the above contains the word "oral" or "spotlight" please see: "oral" presentation means -> notable-top-5% and "spotlight" means -> notable-top-25%. As stated in our emails, we are disassociating presentation type from AC recommendations